# Beyond the Transcript: Translating Non-Coding RNAs and Their Impact on Cellular Regulation

**DOI:** 10.3390/cancers17091555

**Published:** 2025-05-03

**Authors:** Ananya Deshpande, Sagar Mahale, Chandrasekhar Kanduri

**Affiliations:** Department of Medical Biochemistry and Cell Biology, Institute of Biomedicine, University of Gothenburg, SE-40530 Gothenburg, Sweden; ananya.deshpande@gu.se (A.D.); sagar.mahale@gu.se (S.M.)

**Keywords:** peptide ncRNAs, ncRNA-PEPs, peptide-encoding long non-coding RNAs, lncRNA PEPs, lncRNAs, cell cycle, cancer, tumor suppressor genes, oncogenes

## Abstract

Non-coding RNAs (ncRNAs), which constitute a large majority of the human transcriptome, function as regulatory molecules that influence gene expression through specific interactions with DNA, RNA, and proteins. Though ncRNAs are defined by their inability to encode proteins, this review highlights 86 peptides and microproteins encoded by such transcripts. We analyze trends in their activity, consider how conserved they are across species, and discuss their relevance to human health and disease. We explore recent progress in their detection and characterization, providing information on their evolutionary origins, species-specific conservation, and functional contributions to various cellular processes, with particular emphasis on their roles in cell cycle regulation. These findings reflect a shifting paradigm in which non-coding RNAs possess biologically meaningful coding functions, revealing a broad functional repertoire and paving the way for innovative research directions.

## 1. Introduction

During mammalian transcription, a vast array of non-coding transcripts is produced alongside protein-coding ones, comprising approximately 98% of the human transcriptome [1]. Non-coding RNAs (ncRNAs) function as regulatory molecules, influencing gene expression through specific interactions with DNA, RNA, and proteins. Long non-coding RNAs (lncRNAs), typically over 500 nucleotides long, engage with cellular machinery to control diverse biological processes through controlling gene expression at multiple levels: transcriptional, post-transcriptional, and post-translational [2]. In contrast, microRNAs (miRNAs), which are generally 20–25 nucleotides long, primarily function in post-transcriptional gene silencing by binding to messenger RNAs (mRNAs) to inhibit their translation or promote their degradation [3]. These small ncRNAs have garnered significant attention for their broad regulatory roles. The Nobel Prize has been awarded multiple times for RNA research, recognizing the essential role these molecules have played in numerous groundbreaking discoveries throughout history.

While the regulatory functions of lncRNAs have been well-established, recent technological advancements in molecular biology have unveiled a surprising dimension to these transcripts. While lncRNAs were once classified strictly as “non-coding”, recent advances in molecular biology such as ribosome profiling combined with mass spectrometry have challenged this designation [4]. These technologies have revealed that some lncRNAs contain short open reading frames (sORFs) capable of encoding functional peptides, leading to the identification of a new class of biomolecules: ncRNA-encoded peptides (ncRNA-PEPs) and ncRNA-encoded microproteins (ncRNA-MPs). Emerging evidence suggests that these ncRNA-PEP/MPs have versatile functions in cell physiology, influencing processes such as cellular stress responses [5], muscle regeneration [6], immune modulation [7], and cancer biology [8]. These findings underscore the functional breadth of lncRNAs, extending beyond their traditional regulatory roles.

This review focuses on functionally characterized ncRNA-PEPs and ncRNA-MPs and delves into the molecular mechanisms underlying their functions in cellular processes, evolutionary origins, and their broader role in health as well as disease. Considering ncPEPs and ncMPs widely investigated for their role in cancer biology [4,8,9,10,11], we will review the molecular mechanisms by which they influence oncogenic and tumor suppressor functions with a special focus on their role in cell cycle progression. The cell cycle is a highly regulated process crucial for maintaining cellular integrity and tissue homeostasis, with disruptions closely associated with diseases [12]. An increasing body of work suggests that ncRNA-PEPs and ncRNA-MPs actively participate in modulating cell cycle checkpoints and transitions between phases. By reviewing the literature on ncRNA-PEP/MPs and their impact on cell cycle dynamics, we aim to highlight patterns in their regulatory roles across the G_1_, S, G_2_, and M phases, offering insight into how they contribute to the promotion and inhibition of the cell cycle. Their versatile functionality may position ncRNA-PEPs and ncRNA-MPs as valuable therapeutic targets with potential applications in overcoming drug resistance.

### 1.1. Short Open Reading Frames and Translational Potential

To understand the functional roles of ncRNA-PEPs and ncRNA-MPs, it is important to examine short open reading frames (sORFs) within ncRNA transcripts. sORFs are sites that facilitate the translation of peptides generally smaller than 100 amino acids (aa). Historically, strict criteria were applied to predict protein-coding open reading frames to reduce false positives. For instance, coding sequences were required to be at least 100 amino acids long [13], start translation with the standard AUG codon, and exhibit monocistronic transcription [14], meaning each mRNA could only encode a single polypeptide. However, these requirements have since proven insufficient, as many non-canonical ORFs that deviate from these standards have been identified within the human proteome [15]. Over the past two decades, advances in gene expression analyses and novel detection techniques have expanded our understanding of ORFs, revealing that ncRNAs can encode peptides with distinct and sometimes critical biological functions.

### 1.2. Methods to Detect Translating ncRNAs and the Peptides They Encode

Subcellular localization studies, employing reporters like fluorescent proteins [16] and epitope tagging [17], trace the distribution of ncRNAs and the peptides they encode. The cytoplasmic localization of transcripts often serves as an initial indicator of their translation potential [18], which can subsequently be validated through ribosome profiling. Furthermore, analyzing the subcellular localization of ncRNA-PEPs provides valuable insight into their functional roles [19], while mass spectrometry and antibody development facilitate their detection in endogenous contexts and provide an avenue to investigate their interactors.

Large-scale screens integrating Ribo-seq with mass spectrometry have also been employed to identify short ORF-encoded peptides (SEPs). In glioma patients, extracellular vesicles have been analyzed for lncRNA-encoded SEPs [20], while MS-based screens have been utilized to detect lncRNA-derived peptides in response to ionizing radiation [21]. With full-length translating mRNA sequencing and ribosome profiling, Lu et al. (2019) found over 3000 lncRNAs bound to ribosomes, with active translation elongation in widely used cell lines, including HeLa, A549, and Hep3b [22]. However, Chotani et al. (2022) highlight a bias in Ribo-seq studies toward cell lines, noting that the RPFdb database predominantly comprises samples derived from these systems, potentially overlooking sORF translation in primary cells [23]. They validated over 600 novel micropeptides across more than 10 primary cell types and tissues, demonstrating the broader translational activity of lncRNAs beyond the constraints of cell lines. Such screens highlight the abundance of lncRNAs that are translated.

Computational bioinformatics techniques facilitate the efficient identification of SEPs, generating reliable candidate sequences for experimental validation. Zhao and colleagues developed machine learning models to identify plausible sORF candidates from lncRNAs [24]. These models leverage criteria such as sequence features and physicochemical properties unique to SEPs to discover novel micropeptides.

However, these techniques are not without limitations (Table 1). Ribosomes can occasionally translate non-functional sORFs, which may result in false positives when analyzing Ribo-seq data [25]. Each method also has inherent detection biases. Ribo-seq is more likely to detect abundant transcripts, while mass spectrometry often fails to identify peptides present at low abundance [9]. Expression data alone are insufficient to establish the biological relevance of a candidate sORF. Functional validation through assays such as gene knockout, overexpression, or interaction studies is essential to confirm a functional role. Moreover, bioinformatic prediction tools often identify a large number of ORFs that lack evolutionary conservation and show no evidence of cellular function, further complicating interpretation [14].

Considering recent technical advancements and their limitations, it is essential to integrate computational tools, high-throughput screening, and classical molecular biology approaches to identify a previously underexplored class of molecules: translated ncRNAs and their corresponding peptides.

### 1.3. Translational Mechanisms Among ncRNA ORFs

Beyond sORFs, it is also essential to consider the various translation mechanisms of ncRNAs. Like mRNAs, lncRNAs can also undergo canonical RNA processing steps such as 5′ capping with m7G, splicing, and 3′ polyadenylation that facilitates stability and proper localization [26]. While over 90% of mRNAs initiate translation through a cap-dependent mechanism [27], some lncRNAs can initiate translation independently via internal ribosome entry sites (IRES) [28]. This cap-independent translation enables ribosome recruitment and translation initiation under alternative conditions. For instance, certain lncRNAs, like LINC02099, can rely on non-canonical start codons, such as a CUG leucine, instead of the standard AUG methionine to initiate translation [29]. Cai et al. (2021) identified that 53.6% of microproteins they identified in extracellular vesicles of glioma patients had a “non-AUG” start codons [20], providing one explanation for why many microproteins might be missing from MS-based databases.

Another factor that can influence the translation potential of a transcript is a m6A modification, where an adenosine (A) base gains a methyl group at the nitrogen-6 position. m6A is the most prevalent RNA modification, with elevated levels often linked to cancer progression [30]. This modification not only regulates gene expression but is also increasingly recognized as a mechanism that can promote the translation of certain lncRNAs. Notably, lncRNAs that encode peptides such as MRP [31], ATMLP [32], YY1BM [33], and MELOE [28] contain m6A modifications, and these modifications, often in combination with internal ribosome entry site (IRES)-mediated translation, facilitate efficient translation of these lncRNAs.

In some lncRNAs, canonical RNA elements facilitate translation. For instance, in SMIM26 encoding LINC00493, the 3′ poly(A) tail is critical not only for translation initiation but also for maintaining RNA stability [34]. Interestingly, the functional outcomes of certain ncRNA-PEPs can be traced to their translation types, for example, IRES-mediated translation produces more immunogenic peptides than canonically translated peptide from the same MELOE transcript [28]. Taken together, canonical RNA elements, post-transcriptional modifications, alternative start codons, and shorter open reading frames underscore the varied mechanisms of ncRNA translation [35].

## 2. Search Approaches and Data Statistics

### 2.1. Workflow and Data Collection

A systematic search identified 86 functionally characterized peptides encoded by mammalian lncRNA genes, with most reports published after 2020. The search was restricted to functionally characterized peptides encoded from transcripts described as non-coding in the literature. Some transcripts listed below have been re-annotated as protein-coding on the database Ensembl [18,36]. The search targeted reviews and research articles using terms such as lncRNA, non-coding, ncRNA, and miRNA in combination with peptide, microprotein, micropeptide, encoding, non-canonical ORF, sORF, and smORF. We also looked for papers with the phrases ncRNA-encoded peptides, short ORF encoded peptides (SEPs), non-coding RNA encoded peptides (ncPEPs or ncRNA-PEPs), and long non-coding RNA encoded peptides. This effort identified peptides and microproteins encoded by genes traditionally categorized as non-coding, including lncRNAs and pri-miRNAs (Table 2 and Table 3).

A comprehensive list was compiled detailing the gene of origin, transcript classification, publication date, amino acid size, and functional roles of each peptide. The search revealed that 78% of the papers identifying and characterizing ncRNA-PEPs were published on or after 2020 (Figure 1).

### 2.2. Classification of ncRNA-PEP Encoding ncRNAs

There are several types of lncRNAs based on their genomic location relative to nearby coding regions, including intronic, intergenic, antisense, sense, and bidirectional lncRNAs. To assess whether certain lncRNA categories were disproportionately involved in ncRNA-PEP generation, we analyzed the distribution of peptides across these types (Figure 2). Among the total ncRNA transcripts, only 55% was annotated by type. Specifically, peptides were encoded by 5 pri-miRNAs (6%), 18 antisense transcripts (21%), and 24 intergenic transcripts (28%), while 39 lncRNA transcripts (45%) were unannotated.

We found only five characterized miRNA-encoded peptides (miRNA-PEPs) in mammals, but many miRNA-PEPs encoded from 5′ arms of pri-mRNAs have been well-defined in plants. Here, they positively regulate the expression of their own miRNAs in an autoregulatory positive loop [37]. However, in mammals, miRNA-PEPs are less explored, and those identified also exhibit independent effects of the sORF outside miRNA self-regulation [38], such as binding proteins to influence mitochondrial membrane potential [39] or promoting the differentiation of T-regulatory cells to mitigate autoimmune diseases [40].

### 2.3. Size Distribution of ncRNA-PEPs

The identified peptides varied significantly in length, with sizes ranging from 10 to 201 amino acids and a median length of 66 amino acids (Figure 3). As the definitions for “peptides” and “proteins” based on length remain ambiguous, in this report, we broadly classify ncRNA translated products from 1 to 60 amino acids as peptides, 61 to 200 amino acids as microproteins, and more than 200 amino acids as proteins.

**Table 2 cancers-17-01555-t002:** List of ncRNA encoded peptides.

	Peptide	Transcript	Length (aa)	Function	Sources
1	AC115619-22aa	AC115619	22	Prevents assembly of the m6A methyltransferase complex and inhibits HCC progression	[41]
2	AF127577.4-ORF	lncRNA AF127577.4	29	Anti-proliferating function in GBM by reducing both METTL3’s protein stability interaction with ERK2	[42]
3	ASRPS	LINC00908	60	Inhibits STAT3 activation, lowering VEGF expression and suppressing tumor angiogenesis	[43]
4	BVES-AS1-201-50aa	BVES-AS1, C6orf112	50	Enhances CRC invasion ability via Src/mTOR pathway	[44]
5	CIP2A-BP	LINC00665	52	Directly binds to the oncogene CIP2A and inhibits the migration and invasion of TNBC	[45]
6	LINC00665_18aa	LINC00665	18	Modulates the CREB1/RPS6KA3 axis to suppresses the proliferation and migration of osteosarcoma cells	[46]
7	Dleu2-17aa	Dleu2	17	Maintains immune homeostasis by promoting SMAD3 mediated Treg induction	[47]
8	HOXB-AS3 peptide	HOXB-AS3	53	Competitively binds hnRNP A1 and suppresses glucose metabolism reprogramming in CRC cells	[48]
9	IMP	VLDLR	44	Regulates inflammatory gene expression	[49]
10	LINC00339-205-49aa	LINC00339	49	Promotes trophoblast adhesion to endometrial cells and contributes to endometrial receptivity (ER)	[50]
11	LINC00954-ORF polypeptide	LINC00954	49	Promotes pemetrexed sensitivity in drug resistant adenocarcinoma cells	[51]
12	LINC01013-ORF encoded polypeptide	LINC01013	56	Involved in TGFβ1-mediated fibroblast activation	[52]
13	M1 peptide	MALAT1	43	Involved in synaptic function	[53]
14	MAGI2-AS3-ORF5	MAGI2-AS3	45	Anti-tumor role by inhibiting BC cell viability, possibly though interaction with ECM proteins	[54]
15	MELOE-1	meloe	39	Involved in T cell immune surveillance; optimal T cell target for melanoma immunotherapy	[55]
16	MELOE-2	meloe	46	Involved in T cell immune surveillance; optimal T cell target for melanoma immunotherapy	[55]
17	MELOE-3	meloe	54	Expressed in normal melanocytes—not immunogenic like MELOE-1 and MELOE-2	[28,55]
18	MIAC	AQP5-AS1/lncRNA AC025154.2	51	Interacts with AQP2 and inhibits tumor growth and metastasis in HNSCC	[56,57]
19	miPEP-200b	pri-miR-200b	54	Inhibit the EMT of prostate cancer cells by regulating the vimentin mediated pathway	[58]
20	miPEP155	MIR155HG	17	Suppresses autoimmune inflammation by controlling antigen presentation on cells	[7]
21	miPEP31	pri-miRNA-31	44	Represses autoimmunity by promoting Treg differentiation	[40]
22	miPEP497	pri-miR-497	21	miPEP497 does not regulate the levels of its own pri-miRNAs, functions not charecterised	[38]
23	MLN/myoregulin	LINC-RAM/LINC00948	46	Interacts directly with SERCA and regulates muscle performance	[59]
24	MTLN/LEMP/MOXI/SMIM37/MPM/NCRNA00116	LINC00116/1500011K16Rik	56	Multifunctional protein with roles in respiratory efficiency, beta oxidation of fatty acids and myogenesis	[60,61,62]
25	N1DARP	LINC00261	41	Acts as a tumor suppressor and chemosensitizer by regulating USP10-Notch1 oncogenic signaling	[63]
26	PACMP	MARCHF6-DT/CTD-2256P15.2	44	Mediates drug resistance and cancer progression via promoting HR, MMEJ, and PARylation.	[64]
27	pep-AP	lnc-AP	37	Modulates the pentose phosphate pathway and sensitizes CRC cells to Oxaliplatin	[65]
28	PIGBOS	Antisense to PIGB	54	Regulates the unfolded protein response (UPR) in the endoplasmic reticulum and cell death	[66]
29	pTUNAR	TUNAR/LINC00617	48	Regulates neural differentiation and neurite formation by modulating calcium dynamics	[67]
30	PVT1	lncRNA PVT1	10	PVT1 is a tumor antigen recognized by CD8 tumor-infiltrating lymphocytes and mononuclear cells	[68]
31	RNF217-AS1 ORF3-encoded peptide	RNF217-AS1	42	Inhibits macrophage recruitment, pro-inflammatory responses and stomach cancer tumorigenesis	[69]
32	SLT	LINC02099	24	Cytoprotective capacity in cardiomyocytes	[29]
33	SMIM30	LINC00998	59	Promotes the G_1_/S transition of cell cycle by regulating cytosolic calcium level	[70,71]
34	SNHG6 ORF#2	SNHG6	45	Regulates TGF-β/SMAD pathway, influencing endometrial cell development and potentially, gynecological diseases	[72]
35	sPEP1	HNF4A-AS1	51	Facilitates the transcriptional upregulation of stem cell genes related to tumor progression in NB	[73]
36	STMP1/MM47	lncRNA 1810058I24Rik in mice	47	Regulates mitochondrial function to drive Nlrp3 inflammasome activation, cell differentiation, and metastasis.	[74,75,76]
37	STORM	LINC00689	50	Molecular mimic of SRP19, with which it competes for 7S RNA binding (involved in translational regulation)	[77]
38	ELABELA/Toddler	APELA/Gm10664	54	Second endogenous ligand for the Apelin receptor with cardiovascular and developmental roles	[78,79]
39	XBP1SBM	MLLT4-AS1	21	Promotes the expression of VEGF, angiogenesis and metastasis through XBP1s pathway	[80]
40	YY1BM	LINC00278 (Y-linked)	21	Promotes the apoptosis of ESCC cells, m6a modification changes peptide expression in smokers	[33]
41	ZFAS1	Antisense to ZNFX1	57	Promotes higher ROS and HCC cell migration	[81]

**Table 3 cancers-17-01555-t003:** List of ncRNA encoded microproteins.

	Microprotein	Transcript	Length (aa)	Function	Sources
1	115127-microprotein	NONHSAT115127.2	77	Redox stress in extracellular vesicles derived from glioma cancer cells	[20]
2	ACLY-BP	LINC00887	91	Promotes lipid deposition through ACLY stabilization and promotes clear cell renal cell carcinoma	[82]
3	APPLE	ASH1L-AS1	90	Controls eIF4F complex assembly and translation initiation to promote the production of oncoproteins	[83]
4	Arteridin	LncRNA PSR	106	Promotes pathogenic phenotype switching of vascular smooth muscle cells through YBX1 interaction	[84]
5	ASAP	LINC00467	94	Enhances mitochondrial ATP production by stimulating ATP synthase activity and oxygen consumption	[85]
6	ATMLP	AFAP1-AS1	90	Suppresses autolysosome formation and mitophagy to increase tumor cell viability	[32]
7	Aw112010	lncRNA Aw112010	78	Drives IL-12 production and mediates innate immune response	[86]
8	C20orf204-189AA	Linc00176	189	Stabilizes nucleolin and promotes HCC	[87]
9	CASIMO1/SIM22	lncRNA CASIMO1	83	Interacts with squalene epoxidase to influence cell proliferation and cycle progression	[18]
10	CRNDEP	CRNDE	84	Promotes resistance of OvCa cells to treatment with microtubule-targeted cytostatics	[19,88]
11	DDUP	lncRNA CTBP1-DT	186	Contributes to resistance by maintaining Rad18 at damage sites and promoting DNA damage repair	[36]
12	FORCP	LINC00675	79	Regulates apoptosis in response to ER stress to prevent CRC progression	[5]
13	GT3-INCP	LINC00992	120	interacted with GATA3 to coregulate the expression of genes key to the proliferation of ER+ BC cells	[89]
14	HBVPTPAP	lncRNA HBVPTPAP	94	Induces the apoptosis of HCC cells by modulating JAK/STAT signaling pathways	[90]
15	HCP5-132aa	HCP5	132	contributes to adriamycin resistance and regulates ferroptosis to promote BC	[91,92]
16	IGF2-AS-168aa	IGF2-AS	168	Induces trophoblast cell cycle arrest	[93]
17	JunBP	LINC02551	174	Promotes HCC metastasis through c-Jun activation	[94]
18	KRASIM	NCBP2-AS2	99	Decreases KRAS protein levels and the downstream ERK signaling in HCC cells	[95]
19	LINC00511-133aa	LINC00511	133	Promotes the stemness of BC cells via activation of the Wnt/β-catenin pathway	[96]
20	LINC01128-MP	LINC01128	91	Role in intracellular trafficking and endocytosis	[97]
21	Linc013026-68AA	Linc013026/NONHSAT013026.2	68	Role in the proliferation of HCC cells	[98]
22	Lnc-DLX6-AS1 peptide	DLX6-AS1	-	May facilitate NSCLC cell growth by activating the Wnt/β-catenin pathway	[99]
23	MBOP	LINC01234	85	Promoted the expression of MEK1 and activated the MEK1/pERK/MMP2/MMP9 signaling pathway in CRC	[100]
24	MFRLP	LncRNA-MFRL/MSTRG109	64	Inhibits the pathogenic phenotype switch of vascular smooth muscle cells and improves arterial remodeling	[101]
25	miPEP-200a	pri-miR-200a	187	Inhibit the EMT of prostate cancer cells by regulating the vimentin mediated pathway	[58]
26	miPEP133	pri-miR-34a	133	Leads to mitochondrial energy loss and reduced ATP production, inhibiting cancer progression	[39]
27	MRP	LY6E-DT	153	Increases EGFR mRNA stability and transcriptionally activates ZEB1 to promote BC	[31]
28	NBASP/F201A	FAM201A	155	Inhibits neuroblastoma progression by reducing FABP5 expression and inactivating the MAPK pathway	[102]
29	NoBody	LINC01420	68	Regulator of mRNA stability through P-body interaction	[103]
30	pep-AKR1C2	lncAKR1C2	163	Promotes gastric cancer lymph node metastasis by regulating fatty acid metabolism	[104]
31	Pep-KDM4A-AS1	KDM4A-AS1	61	Function in the fatty acid metabolism pathway and reduces ESCC cell viability and migratory capacity	[105]
32	PINT87aa	LINC-PINT (circPINT-exon2)	87	Induces cell cycle arrest and cellular senescence in HCC cells	[106,107]
33	pTINCR/TUBL	TINCR	87	Promotes epithelial differentiation and suppresses tumor growth through CDC42 SUMOylation	[108,109,110]
34	RASON	LINC00673	108	Maintains KRAS in the GTP-bound hyperactive state	[111]
35	RBRP	LINC00266-1	71	Regulates m6A recognition by the m6A reader IGF2BP1 and promotes tumorigenesis	[112]
36	RIP	LncRNA DGCR5	102	Aggravates steroid-induced osteonecrosis of the femoral head (SONFH) in bone marrow mesenchymal stem cells	[113]
37	SMIM26	LINC00493	95	Controls AGK localization and AKT phosphorylation to suppresses CRC metastatic potential	[34]
38	SP0495	KIAA0495, TP73-AS1	201	Tumor suppressive protein that binds to phosphoinositides, promotes autophagy and represses oncogenic signaling	[114]
39	SPAR	LINC00961	90	Regulates Skeletal Muscle Regeneration by inhibiting mTORC1	[6]
40	SRSP	LOC90024	130	Promotes oncogenic mRNA spicing through interaction with splicing regulators	[115]
41	TAT-MIR7-3HG-ORF	MIR7-3HG	125	Protects pancreatic β-cells from dexamethasone induced dysfunction by activating the PI3K/AKT pathway	[116]
42	TP53LC02	lncRNA AC022075.1, FAM169A-AS1	109	Suppresses cell proliferation	[117]
43	TP53LC04	AC022075.1, KLRK1-AS1	100	Suppresses cell proliferation and regulates the cell cycle in response to DNA damage	[117]
44	U9-ORF protein	U90926	87	Secreted by activated myeloid cells, function not fully characterized	[118]
45	UBAP1-AST6	LncRNA UBAP1-AST6	117	Cell proliferation associated function in lung cancer	[22]

## 3. Broad Functional Roles of ncRNA-PEPs and ncRNA-MPs

Non-coding RNA-derived peptides and microproteins have emerged as versatile regulators in both normal physiology and disease. They exhibit an impressive range of functions participating in various cellular processes. In physiological contexts, ncRNA-PEPs and ncRNA-MPs are implicated in immune surveillance [68], muscle contraction [59], synaptic function [53], and cell cycle progression [70]. They are able to act as molecular mimics [77], ligands [79], and antigens [55,68] and influence cellular processes such as protein folding [66], protein trafficking [97], autophagy [32], cytoskeletal maintenance [56,57], ATP production [39,75,85], calcium homeostasis [59], and mRNA stability [103] (Figure 4 and Figure 5).

Peptides encoded by ncRNAs participate in cellular functions in many capacities. Here, we discuss selected ncRNA-PEPs that have been investigated in greater detail for their mechanistic roles in cellular processes. Briefly, AC115619-22aa [41] and AF127577.4-ORF [42] regulate N6-adenosine-methyltransferase assembly and stability, affecting co-transcriptional modifications. AC115619-22aa, encoded by AC115619 lncRNA, suppresses hepatocellular carcinoma progression by interacting with WTAP, disrupting the assembly of the N6-methyladenosine (m6A) methyltransferase complex, and leading to reduced global m6A levels. Similarly, AF127577.4-ORF, encoded by AF127577.4 lncRNA, exerts its tumor-suppressive function by decreasing global m6A levels by reducing the stability of the m6A methyltransferase METTL3 through disruption of the ERK2/METTL3 interaction. Thus, alongside canonical macroproteins, ncRNA-PEPs play critical roles in regulating post-transcriptional modifications.

In metabolic regulation several ncRNA-PEPs have been implicated. MTLN, a 56aa mitochondrial enriched peptide, encoded by a muscle specific lncRNA Linc00116, shown to control fatty acid oxidation [60,61,119]. Interestingly, three independent investigations have demonstrated that MTLN employs distinct mechanisms to regulate fatty acid oxidation, requiring further validation of mechanisms by which MTLN regulate fatty acid oxidation. The micropeptide Stmp1, encoded by the 181-Rik lncRNA, regulates cellular metabolism by maintaining mitochondrial function. It achieves this through modulation of a gene network involving the Nlrp3 inflammasome, the metabolic sensor uncoupling protein 2 (Ucp2), and the calcium sensors S100a8/a9. The micropeptide pep-AP, derived from the lncRNA lnc-AP, mediates its anticancer function by binding to the TALDO1 protein. This interaction inhibits the pentose phosphate pathway (PPP), thereby decreasing NADPH/NADP^+^ and glutathione (GSH) levels, which in turn promotes the accumulation of reactive oxygen species (ROS) and triggers apoptosis [65]. Additionally, the HOXB-AS3 peptide is a conserved 53-aa peptide, encoded by a lncRNA HOXB-AS3, which exerts its anticancer functions by disrupting hnRNP A1-mediated splicing of pyruvate kinase M (PKM), resulting in lower PKM and suppressing glucose metabolism reprogramming [48]. These ncRNA-PEPs collectively influence energy homeostasis (Figure 4). Their ability to engage with established protein networks opens new avenues for targeting cancer metabolism, oxidative stress, and mitochondrial dysfunction.

Interestingly, ncRNA-PEPs can function as signaling entities themselves or regulate the expression of other signaling molecules. The annotated non-coding transcript AK092578 was predicted to encode a conserved 32aa hormone Elabela. Subsequent functional analyses established Elabela as an endogenous ligand for the *apelin receptor* (*Aplnr*), with the Elabela–Aplnr signaling axis playing a pivotal role in early cardiovascular development [78]. XBP1s-binding micropeptide (XBP1SBM) is a 21-amino acid peptide encoded by *MLLT4-AS1*. It promotes angiogenesis in cancer cells by enhancing the nuclear localization of XBP1s, leading to the activation of the key angiogenic factor VEGF [80].

ncRNA-PEPs also support the cell’s stress responses. PIGBOS1-encoded 54aa PIGBOS participates in the unfolded protein response to maintain proteostasis under genotoxic stress [66]. CTD-2256P15.2-encoded PACMP supports DNA damage responses through its dual function to maintain CtIP abundance and promote poly(ADP-ribosyl)ation [64]. ncRNA-PEPs like MLN, pTUNAR, and SMIM30 have been implicated in the modulation of intracellular calcium levels by influencing the calcium ATPase SERCA. In immune regulation, miPEP155 enhances antigen presentation, while MELOE peptides and PVT1 function as antigens themselves. Finally, MIAC inhibits actin filament formation, shaping cytoskeletal integrity.

This remarkable functional versatility is not confined to ncRNA-PEPs but extends to ncRNA-MPs as well. LncRNAs, which harbor sORFs, have long been suspected of encoding peptides, but the extensive list of functionally characterized microproteins has been surprising. These microproteins influence cellular processes such as ATP generation (e.g., ASAP [85], miPEP133 [39]) and ROS accumulation (e.g., MFRLP [101]). They also influence autophagy (SP0495 [114] and PINT87aa [107]) and autolysosome formation (ATMLP [32]). In post-transcriptional modifications, SRSP modulates mRNA splicing [115], while NoBody contributes to p-body dissolution [103] (Figure 5).

In translation regulation, ncRNA-MPs play pivotal roles: RBRP influences m6A recognition by the m6A reader IGF2BP1 [112], and APPLE enhances translation initiation [83]. Their impact extends to signaling pathways, where KRASIM reduces GTPase KRAS levels [95] and RASON stabilizes KRAS in its hyperactive, GTP-bound state [111]. In the Wnt pathway, RIP inhibits nuclear translocation of β-catenin [113], whereas LINC00511-133aa promotes it [96]. Furthermore, ncRNA-MPs affect lipid metabolism, with CASMIO1 facilitating lipid droplet clustering [18] and ACLY-BP stabilizing ATP citrate lyase (ACLY) [82], thereby promoting lipid deposition. Stress responses are similarly influenced: FORCP manages ER stress and prevents apoptosis [5], HCP5-132aa inhibits ferroptosis [91], and TP53LC04 and DDUP participate in the DNA damage response [36,117]. CRNDEP associates with centrosomes during cell division [19], highlighting the potential role of ncRNA-MPs in mitotic processes. Structural and intercellular processes are also regulated by ncRNA-MPs. For example, pTINCR organizes cortical actin and strengthens cell-to-cell adhesion [110], while LINC01128-MP participates in intracellular trafficking and endocytosis [97]. Aw112010 induces IL-12p40 production [86], and 115127-microprotein is found in extracellular vesicles [20], suggesting potential roles in intercellular communication. Although ncRNA-PEPs and ncRNA-MPs are classified based on their length, a review of the literature suggests no clear functional divergence between the two. However, ncRNA-PEPs appear to have distinct roles in tumor immune evasion, particularly through the regulation of antigen presentation [55,68]. Given their strong association with anti-tumor immune responses, they are being proposed as potential candidates for cancer vaccines [120].

The roles of ncRNA PEP/MPs extend to both protective and pathological responses in various diseases, including diabetes [116], osteonecrosis [113], and cardiovascular disorders [121]. Notably, approximately 66% of the published ncRNA-PEP/MPs identified in our search have been investigated for their roles in cancer. In cancer, ncRNA-PEPs and ncRNA-MPs participate in several key processes, modulating hallmark traits such as angiogenesis [43], metastasis [34], proliferation [18], metabolic reprogramming [48], and DNA damage repair [117]. Their roles in cancer have been thoroughly reviewed in the recent years [4,8,9,10,11].

In this review, we focus particularly on their role in cell cycle regulation, where these molecules demonstrate a dual capacity: promoting tumorigenesis by disrupting checkpoint control or exerting protective effects by inducing cell cycle arrest under stress. This functional dichotomy highlights the complexity of their contributions to cancer biology.

Thus far, we have provided an up-to-date overview of ncRNA-derived peptides and microproteins and their diverse cellular functions. Building on this foundation, we now turn to a relatively underexplored aspect of the field—their potential involvement in cell cycle control. Given the central role of the cell cycle in development, tissue homeostasis, and disease, elucidating how these molecules influence its regulation is a critical and timely research direction. We summarize current findings, propose mechanistic links, and outline key gaps to inform future investigations.

## 4. ncRNA-PEP/MPs in Cell Cycle Regulation

The cell cycle is a tightly regulated series of steps that cells undergo to grow and divide, ensuring accurate DNA replication and division of cellular contents. Here, we delve into ncRNA-PEPs and ncRNA-MPs implicated in cell cycle progression. While only a few peptides/microproteins are directly involved, many show potential regulatory roles, with some indirectly influencing progression. Of the 86 peptides/microproteins identified, 15 were found to impact the cell cycle (Table 4). In the following section, we explore their functions across different phases (Figure 6).

### 4.1. The Potential Role of ncRNA-PEP/MPs in G_1_ Phase of the Cell Cycle

The G_1_ phase is crucial for cell growth and preparation for DNA synthesis. Key regulators include cyclins and CDKs, which drive the cell past the G_1_ checkpoint by promoting the phosphorylation of the retinoblastoma (Rb) protein [123]. This phosphorylation releases E2F transcription factors, enabling the transcription of genes essential for DNA replication and progression to the S phase (Figure 7).

#### 4.1.1. ncRNA-PEP/MPs in G_1_/S Transition: Modulation of Transcription Factors, Cyclins, and CDKs

NcRNA-PEPs and ncRNA-MPs are involved in modulating transcription factors and CDK/cyclin complexes important for the restriction checkpoint. The 133aa microprotein miPEP133 is encoded from the pri-miRNA miR-34a. It interacts with the mitochondrial chaperon HSPA9 to reduce mitochondrial membrane potential and ATP production, leading to cell cycle arrest at G1. It functions as a part of the p53 response, being transcriptionally activated by wildtype p53 while also promoting the transcriptional activation of p53 [39].

Phosphorylation of Rb, a critical step in determining the G_1_/S transition [123], is regulated by the Cyclin D-CDK4/6 complex in early G_1_ and the Cyclin E-CDK2 complex in late G_1_ (Figure 7). Two ncRNA-PEPs, STMP1 and SMIM30, have been identified as regulators of cyclin E levels during this transition [70,75]. STMP1, initially identified in mice as MM47 and encoded by the lncRNA 1810058I24Rik [76], plays a crucial role in the G_1_/S transition [73]. Sang and collegaues show that silencing STMP1 impedes this transition by reducing ATP production, as STMP1 enhances mitochondrial complex IV activity [75]. This energy boost supports elevated Cyclin E/CDK2 levels, facilitating G_1_ progression. Similarly, SMIM30, a 59aa peptide encoded by LINC00998, has also been shown to promote cell proliferation by enhancing G_1_/S transition through the Rb pathway [70]. Yang et al. (2023) find that silencing SMIM30 decreased the protein levels of CDK4, cyclin E2, and E2F1, as well as phosphorylated Rb. Furthermore, they show a decrease in the mRNA levels of E2F1 target genes following SMIM30 silencing, strongly suggesting that ncRNA-MP SMIM30 promotes the G_1_/S transition by regulating the cyclin/CDK-Rb-E2F1 pathway.

#### 4.1.2. Potential Role of ncRNA-PEPs and ncRNA-MPs in G_1_/S Transition via Metabolic Homeostasis

In addition to modulating cyclins and CDKs, ncRNA-MPs might also be capable of influencing metabolic homeostasis, which is a critical factor for cells at the restriction checkpoint. As discussed above, SMIM30 promotes G_1_/S transition, and interestingly, Yang et al. (2022) posit that this effect may be mediated by SMIM30 enhancing the activity of the calcium ATPase SERCA, thereby lowering cytosolic calcium levels [70]. This mechanism aligns with SMIM30’s primary localization to the ER, an organelle essential for maintaining cytosolic calcium homeostasis. Since calcium regulation is a crucial element of cell cycle progression [124] and it is significant that ncRNA-MPs could actively contribute to this process.

Expanding beyond calcium homeostasis, lipid metabolism emerges as another critical facet of metabolic regulation during the G1/S transition. The micropeptide CASIMO1, encoded by the lncRNA CASIMO1, modulates lipid metabolism and promotes the G_1_/S transition in breast cancer cells [18]. CASIMO1 positively regulates squalene epoxidase (SQLE), an enzyme essential for cholesterol biosynthesis. These lipids are crucial not only for membrane formation but also as signaling molecules that regulate cellular proliferation, particularly during the G_1_/S transition in rapidly dividing cells. This role in lipid homeostasis could be why loss of CASIMO1 results in a G_0_/G_1_ arrest.

#### 4.1.3. Linking ncRNA-MPs to G_1_/S Progression via Autophagy

Beyond metabolic regulation, ncRNA-MPs play crucial roles in autophagy. Two ncRNA-MPs, PINT87aa and SP0495, induce G_1_ arrest upon overexpression [107,114]. Interestingly, while both regulate autophagy, they do so in opposite ways but converge on the same functional outcome of G_1_ arrest. PINT87aa, encoded by a circular form of the intergenic lncRNA LINC-PINT, inhibits mitophagy via the PINK1/Parkin axis [107]. Through FOXM1 interaction, PINT87aa downregulates prohibitins, which govern mitophagy, thereby reducing mitophagic activity. SP0495, a TP73-AS1-encoded peptide, binds phosphoinositides and modulates the stability of key autophagy regulators to *induce* autophagy [114]. Using a cycloheximide (CHX) assay, Li et al. (2023) demonstrated that SP0495 ectopic expression significantly extended the half-life of BECN1, a core component of the autophagy initiation complex [125]. SP0495 is the largest ncRNA-MP identified in our search (201-aa), with both transmembrane and signal peptide domains, and localizes to the Golgi apparatus [114]. The Golgi functions as a hub for vesicle formation and lipid supply, essential for autophagosome biogenesis [126]. In line with this, SP0495 was shown to induce autophagosome accumulation in cancer cells [114].

While both studies demonstrate that PINT87aa and SP0495 induce G_1_ arrest and are involved in autophagy, neither link autophagy directly to the cell cycle arrest. However, Xiang et al. (2021) suggest a potential link [107]. They report that impaired mitophagy leads to the accumulation of dysfunctional mitochondria in senescent cells. The link between autophagy and cell cycle progression has been explored in the past [127]. Increased autophagy can facilitate cell cycle progression by degrading damaged organelles and proteins, improving cellular health, and maintaining energy balance. This process supports a smooth G_1_ transition by supplying essential building blocks and maintaining metabolic equilibrium. However, excessive autophagy can trigger autophagic cell death or senescence [128], emphasizing the need for balance.

Thus, the ncRNA-MPs PINT87aa and SP0495 regulate both ends of this spectrum. While PINT87aa inhibits mitophagy, SP0495 promotes autophagy, highlighting the nuanced roles of ncRNA-MPs in autophagy regulation. Elucidating the molecular mechanisms linking autophagy to ncRNA-MP-mediated G_1_ arrest and senescence remains an interesting area for further exploration.

### 4.2. The Potential Role of ncRNA-PEP/MPs in S Phase

#### 4.2.1. ncRNA-PEPs in S Phase: PCNA Modulation

The S phase is the part of the cell cycle where DNA replication occurs, resulting in the duplication of the cell’s genetic material to prepare for cell division. Proliferating Cell Nuclear Antigen (PCNA) functions as a sliding clamp during the S phase, enhancing the processivity of DNA polymerases, thereby facilitating efficient and accurate DNA replication [129]. Interestingly, two lncRNA-encoded peptides exert opposing effects on PCNA levels. BVES-AS1-201-50aa is a peptide encoded from the transcript BVES-AS [44], which promotes PCNA expression in colorectal cancer cells, while LINC00954-ORF polypeptide, a 49aa peptide identified very recently, suppresses PCNA levels in pulmonary adenocarcinoma cells [51]. Authors show the change in PCNA levels by Western blotting and IF, as a marker for modulated cell proliferation. However, elucidating the mechanistic details of how the ncRNA-PEPs work will help us understand if their influence on S phase progression is by direct interaction with key players or an indirect consequence of their roles parallel to the cell cycle.

#### 4.2.2. ncRNA-PEPs in S Phase: DNA Damage Repair

Beyond modulating PCNA levels, ncRNA-PEP/MPs also play a critical role in DNA damage repair. While NHEJ is the preferred repair mechanism in G_1_, homologous recombination is utilized during replication in the presence of the sister chromatid to ensure accurate damage repair [130]. Upon DNA damage, the polysome association of the lncRNA CTBP1-DT increases, leading to the production of the microprotein **DDUP** (DNA damage-upregulated protein). DDUP is subsequently phosphorylated by ATR, triggering a conformational change that promotes the retention of Rad18 at damage sites [36]. Through interactions with Rad51C and PCNA, DDUP facilitates damage resolution via homologous recombination and post-replication repair, respectively. It is intriguing that lncRNAs, once thought to be non-coding, can produce microproteins large enough to have secondary structures and can be regulated by post-translation modification-mediated conformational changes. Although DDUP is relatively large (186aa), peptides from smaller sORFs have also been shown to undergo similar post-translational modifications [131]. Overall, these findings underscore the potential role of ncRNA-PEP/MPs in regulating key aspects of the S phase, from DNA replication to damage repair.

### 4.3. The Potential Role of ncRNA-PEP/MPs in G_2_/M

The G_2_ stage of the cell cycle is the final phase before mitosis, during which the cell undergoes critical preparations, including the synthesis of proteins and organelles necessary for cell division [132]. Additionally, the cell conducts quality control checks to confirm that DNA replication has occurred without errors, ensuring genomic integrity before entering the mitotic phase. At the G_2_/M checkpoint, regulatory molecules work to either halt damaged cells or enable cell division.

CDK1 is particularly essential for the G_2_/M transition [133]. When bound with cyclins A and B, CDK1 phosphorylates various proteins to initiate mitosis, leading to chromatin condensation, nuclear envelope breakdown, and spindle formation. Interestingly, the LINC00954-ORF polypeptide, which has been discussed in the context of the S phase, is upregulated and leads to a reduction in CDK1 levels [51]. MIAC (Micropeptide Inhibiting Actin Cytoskeleton), a peptide translated from AQP5-AS1, directly binds to AQP2 and inhibits renal cell carcinoma growth by inhibiting PI3K/AKT and MAPK signaling downstream of EREG/EGFR signaling [56]. Notably, the overexpression of MIAC results in cell cycle arrest at S and G_2_ phases, as observed with flow cytometry, suggesting its involvement in G2 to M progression. However, the precise mechanisms through which MIAC regulates this transition are yet to be elucidated. Interestingly, a microprotein encoded from KLRK1-AS1, TP53LC04, has been clearly implicated in cell cycle regulation. Unlike the other two microproteins, its function in cell cycle control is well characterized, as it induces G2/M arrest as part of the p53-dependent damage response [117].

As cell cycle regulators, ncRNA-MPs also govern membrane-less organelles during the G_2_/M transition. P-bodies, or processing bodies, are specialized cellular structures where mRNA molecules are stored, degraded, or processed. These granules contain enzymes and proteins involved in mRNA decay, translation repression, and RNA interference, helping to regulate gene expression by controlling which mRNAs are available for translation. P-bodies form in G_1_ and progressively enlarge during interphase, particularly in G_2_, and dissolve in mitosis [134,135]. Interestingly, among the many regulatory proteins encoded by P-body mRNAs, cell-cycle regulators such as cohesin and condensin were shown to be particularly enriched [136], making the timely formation and dissolution of P-bodies important for cell cycle progression

The long non-coding RNA LINC01420 encodes a 68aa microprotein known as NoBody, which plays a vital role in regulating P-bodies [103]. In vitro, NoBody undergoes liquid–liquid phase separation in the presence of RNA. Phosphorylation of NoBody at the G_2_/M checkpoint by CDKs modulates these phase separation properties and promotes their dissociation during mitosis [137], underscoring the critical role of NoBody in controlling the dynamics of P-body formation as the cell progresses through the cycle. Taken together, these findings highlight the role of ncRNA-PEP/MPs in orchestrating the events of the G_2_/M transition to maintain proper cell cycle progression, with both direct well-studied roles as well as indirect potential roles.

### 4.4. The Potential Role of ncRNA-PEP/MPs in Cell Division

Following G_2_, cells enter mitosis where DNA is condensed and chromosomes are segregated, dividing the cell into two genetically identical daughter cells. Spindle fibers pull chromosomes apart, and cellular structures reorganize to support this division. A few ncRNA-MPs show indirect roles in involved in microtubule organization and protein trafficking in relation to this division.

CRNDEP, an 84aa long microprotein, is encoded by LINC00180 and exhibits distinct localization patterns depending on the cell cycle phase [19]. It shifts from the nuclear matrix in early G_1_ to the nucleoli in late G_1_ and S, and finally to the centrosomes during mitosis. This dynamic distribution suggests a role in coordinating cell cycle progression. Mass spectrometry analysis identified interactions between CRNDEP and proteins such as NuMA, Plk1, and Nlp, which are involved in centrosome maturation and mitotic spindle formation, underscoring its potential involvement in mitotic regulation. Exploring these identified interactions may provide further insights into CRNDEP’s functional mechanisms. Notably, in ovarian cancer cell lines where CRNDEP is silenced, cells exhibit reduced division rates compared to controls. While the exact mechanistic details of CRNDEP’s activity is yet to be elucidated, taken together, the data above make it hard to ignore its role in mitosis.

Finally, two ubiquitin-like ncRNA-MPs: pTINCR and Ub^KEKS^ have also been linked to cell cycle regulation, though their roles are not linked to specific stages of the cycle. pTINCR, which is encoded by the lncRNA TINCR [108] is upregulated by p53 in response to cellular damage and induces transcriptional changes associated with the actin cytoskeleton and cell cycle regulation [110]. pTINCR expression was shown to be inversely correlated with cell cycle genes; however, its role in the cell cycle beyond this is yet to be explored. However, Ub^KEKS^, which is encoded from a pseudogene, UBBP4 [138], is somewhat better characterized in this context. Ribosome profiling has revealed that several long non-coding RNAs derived from pseudogenes serve as significant sources of ORFs. UBBP4 serves as a protein coding example, as it produces a unique ubiquitin variant, which differs from standard ubiquitin by four amino acids: Q2K, K33E, Q49K, and N60S. This modified variant has distinct protein targets compared to canonical ubiquitin, playing a specialized role in regulating protein trafficking between the nucleolus and nucleoplasm [122]. This process is essential for the localization of cell cycle-related proteins like structural lamins that support the nuclear envelope and control mitotic regulation. Frion and colleagues show that Ub^KEKS^ knockout alters nucleolar protein composition, notably affecting proteins linked to cell cycle control and stress responses [122]. They observed the nucleolar sequestration of IFI16 and p14ARF, which are important for sensing DNA damage and inducing cell cycle arrest. This serves as an example of a lncRNA derived from a pseudogene with the ability to code for a protein with a potential role in cell division.

In summary, ncRNA-PEP/MPs are capable of exerting diverse and essential roles across each stage of the cell cycle. Having reviewed the functional roles of these peptides through the cell cycle stages, we now turn to a mechanistic analysis of how these selected ncRNA-PEP/MPs accomplish their functions at the molecular level.

## 5. Mechanisms of ncRNA-PEP/MPs

### Layers of Regulation

**ncRNA-PEP/MPs at the transcriptional level regulation**: ncRNA-encoded peptides and microproteins exert cell cycle control through various mechanisms, working across transcriptional, post-transcriptional, and post-translational levels (Figure 8). Transcriptionally, miPEP133, from the primary miR-34a transcript, strengthens its own transcription in a positive feedback loop, and enhances the expression of p53-regulated genes such as FAS, MDM2, PUMA, and P21 [39].

**NcRNA-PEP/MPs at the post-transcriptional level regulation**: Beyond transcription, microproteins like CRNDEP impact the cell cycle’s progression by interacting with RNA-processing enzymes, such as the helicase p54, which modulates RNA post-transcriptionally and is crucial for G_2_/M transition [19]. Another ncRNA-MP exerting control at the post-transcriptional level is NoBody, encoded by LINC01420, which regulates P-body dynamics to manage mRNA decay and stability [103]. As a consequence, it impacts mRNA availability and translation rates—a vital role for controlling gene expression timing.

**NcRNA-PEP/MPs at the post-translation level regulation**: In the post-translational landscape, the unconventional ubiquitin variant Ub^KEKS^ modifies cell cycle proteins to influence their localization rather than marking them for degradation [122]. The ncRNA-MP pTINCR is able to facilitate sumoylation to modulate Cdc42 activity to influence epithelial differentiation and tumor suppression [110].

**NcRNA-PEP/MPs as biomolecule interactors**: Many ncRNA-PEP/MPs like MIAC and PINT87aa have the ability to bind other proteins to modulate their functions, as confirmed by Co-IP experiments. MIAC directly binds the protein AQP2 protein and inhibits the activation of the EGFR signaling pathway [56], while PINT87aa controls cellular senescence induction by FOXM1 binding [107]. As diverse regulators, ncRNA-PEP/MPs’ interaction is not limited to proteins but extends to other biomolecules like lipids. Autophagy, a process driven by membrane dynamics, relies heavily on lipids for its regulation [139]. The ncRNA-MP SP0495 has been shown to bind phosphoinositides, particularly PI(3)P and PI(3,5)P2, and to inhibit AKT phosphorylation [114] with protein–lipid overlay assays. AKT undergoes a conformational change that allows for phosphorylation and activation upon binding phosphoinositides on the inner plasma membrane. Li and colleagues hypothesize that SP0495 may regulate autophagy and AKT phosphorylation through its interaction with specific phosphoinositides, a theory they suggest could be further tested through liposome flotation assays [114]. It is intriguing that this novel class of microproteins can modulate well-characterized central signaling players such as AKT.

Additionally, some ncRNA-MPs like DDUP act as scaffolds (Figure 8). DDUP enhances the Rad18–Rad51c interaction to facilitate homologous recombination repair and Rad18–PCNA interaction to facilitate post-replication repair [36]. Collectively, these examples reflect the multifaceted ways ncRNA-derived peptides and microproteins could control cellular processes, demonstrating a complex, layered network of regulation that could be critical to homeostasis and proliferation.

## 6. Evolutionary Conservation of Cell Cycle Regulating ncRNA-PEP/MPs

To see if these peptides/microproteins are conserved and if this type of regulation originates with an early common ancestor, we conducted a conservation analysis. The analysis was designed to assess sequence conservation, with an identity threshold set at over 85%. In BLAST (version 2.16.0), the Expect value (E-value) indicates the probability that a match occurred by chance; a lower E-value threshold suggests greater confidence in the significance of the sequence similarity. To ensure relevance, especially considering the short sequence lengths, we applied a cutoff of E-value ≤ 1 × 10^−10^, filtering out sequences that were unlikely to be related.

The presence of sORFs has historically been disregarded, under the assumption that they lack biological significance. However, the recent characterization of sORFs and their involvement in cellular processes makes investigation of their evolutionary origins increasingly relevant. In a comprehensive analysis by Sandmann et al. (2023), the evolutionary origins and conservation mechanisms of 7264 human sORFs were investigated, revealing that almost 90% of these sORFs are evolutionarily young, with 4101 emerging de novo from ancestral non-coding regions [97].

De novo-evolved proteins are newly evolved proteins that originated from previously non-coding regions of the genome rather than through modification or duplication of existing protein-coding genes [140]. Ruiz-Orera et al., (2018) suggest that the chance occurrence of ORFs with a favorable codon composition facilitates the translation of these neutrally evolving peptides [141].

Many de novo proteins, particularly those in humans, have evolved relatively recently; for instance, within primates. This rapid emergence suggests that they may contribute to species-specific traits and adaptations. According to our BLAST search (Figure 9), many cell cycle-regulating microproteins discussed in this paper, such as CRNDEP, miRNA-PEP133, CASIMO1, SP0495, PINT87aa, and DDUP are exclusive to primates. pTINCR and NoBody exhibit conservation within placental mammals, with some, like STMP1 and SMIM30, extending across therian mammals, indicating broader functional roles. However, certain peptides/microproteins appear to be unique not only to primates, but specifically to humans such as MIAC, BVES-AS1-201-50aa, and TP53LC04, underscoring the potential roles of de novo proteins in human-specific traits.

Despite their recent origins, de novo proteins can integrate into existing cellular pathways, where they establish interactions with other proteins. This rapid integration is likely facilitated by their typically short and disordered structures, which may provide the flexibility needed for binding with more established cellular components [142]. Interestingly, the ncRNA-PEP/MPs that we have looked at in this review appear limited to mammals or even primates but function within highly conserved signaling pathways like MAPK [71] and Wnt/β-catenin [114]. They also interact with highly conserved proteins, such as ATR [36] that is preserved across all eukaryotes [143]. An interesting hypothesis is that these newly evolved ncRNA-PEP/MPs are meeting emerging cellular needs while integrating into long-standing regulatory networks. This species-specific presence of ncRNA-PEP/MPs that interact with highly conserved proteins may be linked to organismal complexity.

However, assessing the degree of conservation of these sORFs can be challenging. Patraquim et al. (2022) argue that standard homology detection with BLAST, based solely on sequence similarity, is insufficient for identifying sORF orthologues [144]. To address this, they developed bioinformatic tools that use iterative “jackhammer” searches, along with reciprocal searches and MAFFT alignment scores, to enhance the accuracy of sORF homolog detection. They employ this pipeline “GENOR” in *Drosophila*, where they detect homologues for 186 out of 191 proteins in Flybase that previously had no annotated homologies, highlighting that the current standards for assessing conservation could be restrictive.

## 7. NcRNA-PEP/MPs in Cancer Biology

### 7.1. Role of Cell Cycle Related ncRNA-PEP/MPs in Physiology and Disease Physiology

NcRNA-PEP/MPs that could influence the cell cycle have primarily been identified through cancer research, though some are also involved in normal physiological processes. For example, NoBody is essential across various cell lines, where it regulates the mRNA decapping complex and P-body dissolution [103]. This role is key to maintaining cellular homeostasis and extends beyond cancer-related functions. Similarly, Ub^KEKS^ participates in crucial post-translational modifications [138] in normal cellular physiology, suggesting functions for ncRNA-PEPs beyond pathological states.

Interestingly, some authors propose that even ncRNA-PEP/MPs that have been studied primarily with a lens of cancer, might have a role beyond that. For example, the decrease in proliferation upon loss of CASIMO1 is consistent in multiple breast cell lines (MCF7, KPL1, and T47D), including non-transformed cells [18]. This impairment of cell proliferation upon CASIMO1 knockdown in non-transformed MCF10a cells suggests that CASIMO1’s function is relevant in physiological contexts as well.

The lncRNA TINCR has been implicated in epithelial cell differentiation [145]. It has recently been posited that rather than the RNA itself, the microprotein it encodes (pTINCR) is crucial for mediating these roles and ensuring skin homeostasis [108,110]. However, beyond its physiological functions, pTINCR is also implicated in epithelial cancers [109]. This dual role in both cellular homeostasis and cancer suggests that ncRNA-MPs could have underexplored functions outside of tumorigenesis.

### 7.2. Cell Cycle Related ncRNA-PEP/MPs as Oncogenes and Tumor Suppressors

**Oncogenic ncRNA-PEP/MPs**: ncRNA-PEP/MPs involved in the cell cycle show consistent themes in oncogenesis and tumor suppression across various cancers, which is not surprising considering that they are able to both promote and inhibit the progression of the cell cycle. While we discussed the role of a few ncRNA-MPs in physiology, the majority of the ncRNA-PEP/MPs we have analyzed in relation to the cell cycle are primarily studied in the context of cancer. These peptides/microproteins often drive malignancies by bypassing natural growth barriers, facilitating unchecked division across multiple cancer types. For example. The oncogenic ncRNA-PEP BVES-AS1-201-50aa enhances proliferative signaling pathways, driving rapid growth seen in CRC [44]. ncRNA-PEP/MPs can also influence metabolic reprogramming or stress response pathways, promoting cell growth in environments that might otherwise be restrictive. For instance, SMIM30 [70], CASIMO1 [18], and STMP1 [75] increase metabolic capacity to fuel cell proliferation, while DDUP supports DNA repair mechanisms [36], helping cells endure replication stress common in malignancies.

**Tumor suppressor ncRNA-PEP/MPs**: In contrast to oncogenic ncRNA-PEP/MPs, tumor-suppressive ncRNA-MPs like miRNA-PEP133, SP0495, and TP53LC04 enforce strict regulatory controls that maintain cell cycle checkpoints and induce apoptosis in response to genomic stress [39,114,117], often through p53-related pathways. These tumor suppressors are vital for preventing oncogenesis by preserving DNA integrity and managing cellular damage. For example, miRNA-PEP133 upregulates p53, triggering a cascade of anti-tumor effects, including apoptosis and cell cycle arrest [39]. Meanwhile, PINT87aa regulates autophagy to limit energy availability in potentially oncogenic cells [107], stabilizing cell growth and preventing tumor formation.

Along with PINT87aa, functional studies have implicated the ncRNA-PEP MIAC as a tumor suppressor. Knocking out PINT87aa in Hs683 glioma cells increased xenograft tumor volumes. To a similar effect, the depletion of MIAC generated a higher number of lung metastatic nodules in vivo compared to the control [56]. This dual capability of ncRNA-PEP/MPs, where they can either support or inhibit tumor growth, emphasizes their versatile role in cancer biology.

The downregulation of tumor suppressor ncRNA-MPs in cancer raises an important point about their unexplored roles in normal physiological processes. For instance, SP0495 is silenced in numerous primary tumors due to promoter methylation of its transcript TP73-AS1. This modification is absent in normal tissues [114], suggesting that SP0495 has influential functions under normal cellular circumstances. Many ncRNA-PEP/MPs, often studied for their tumor-suppressive roles, may also be essential for cellular homeostasis and tissue health. The role of ncRNA-PEP/MPs outside of oncogenesis and in maintaining physiological balance is a sparsely explored but potentially impactful avenue for scientific exploration, one that could give us insights into their broader biological functions.

### 7.3. ncRNA-PEP/MPs in Drug Resistance

The treatment of genotoxic-resistant cancers remains a significant challenge in oncology [146]. The heightened ability of cancer cells to repair DNA damage induced by therapeutic agents contributes to chemoresistance across multiple cancer types, often resulting in tumor progression and relapse. If we look at ncRNA-MP involvement in this process, DDUP is upregulated in response to cisplatin treatment, facilitating DNA repair and thereby enhancing resistance to platinum-based chemotherapy [36]. In contrast, LINC00954-ORF polypeptide has been shown to sensitize cancer cells to pemetrexed [51], another chemotherapeutic agent, highlighting its therapeutic potential to counteract resistance. In a similar vein, Li et al. (2020) show that the chemically synthesized MIAC polypeptide can inhibit renal cancer growth both in vitro and in vivo [56], underscoring the potential of ncRNA-PEPs as treatment strategies.

Furthermore, CRNDE(P) overexpression has been shown to enhance cellular resistance to various microtubule-targeting chemotherapeutic agents, such as paclitaxel and nocodazole, by accelerating microtubule polymerization and promoting cell survival [19]. This resistance appears closely tied to CRNDE(P)’s role in stabilizing microtubule structures, which undermines the efficacy of drugs that typically work by disrupting these structures to arrest the cell cycle. Understanding the mechanisms of ncRNA-MPs is, thus, of substantial clinical relevance. For example, in CRNDE(P)-overexpressing cells, combining paclitaxel with noscapine (NOS), another microtubule stabilizer, counteracts this resistance by increasing the impact of the drugs on microtubule function, suggesting a potential therapeutic strategy to overcome CRNDE(P)-mediated chemotherapy resistance in ovarian cancer.

In summary, ncRNA-PEP/MPs play complex and dual roles in both oncogenesis and physiological processes, influencing cell cycle regulation, DNA repair, and cellular homeostasis. The exploration of these peptides/microprotein in cancer and beyond not only deepens our understanding of tumor biology but also opens new avenues for targeted therapeutic and biomarker development, potentially improving cancer treatment precision and efficacy.

### 7.4. Clinical Relevance of Cell Cycle Related ncRNA-PEP/MPs

The cell cycle is a critical target for cancer therapeutics, with various strategies aimed at inhibiting microtubule activity, topoisomerases, or checkpoint kinases [12]. However, targeting the cell cycle presents a significant challenge, as these therapies can also affect healthy cells. For example, CDK inhibitors impair cell division in rapidly proliferating cells and can also suppress immune function in the bone marrow, causing immunosuppression, and disrupting the high-turnover cells of the gastrointestinal system, leading to digestive complications [147]. Therefore, there is a pressing need for more targeted approaches in cancer therapy. Long non-coding RNAs have gained attention due to their tissue-specific expression patterns, which may allow for selective targeting of disease-associated cells [148], a benefit that would extend to the peptides they encode.

## 8. Future Perspectives

LncRNAs, traditionally characterized by their non-coding nature, have not been thoroughly explored for their potential coding roles. They harbor sORFs capable of encoding peptides, as well as numerous microproteins that possess secondary structures and can undergo conformational changes. The investigation of ncRNA-PEP/MPs in the context of cell cycle progression has highlighted their potential as versatile modulators of various stages within this highly regulated process. Our analysis identified 15 ncRNA-PEP/MPs with roles linked to the cell cycle, but mechanistic characterization of their functions in this context remains largely absent. Taking a step back to consider the broader implications of this report, key gaps persist that hinder a comprehensive understanding of ncRNA-PEP/MPs.

Looking at ncRNA-PEP/MPs in contexts beyond oncogenesis, especially in regulating physiological stability, is a relatively untapped yet promising area of research that could shed light on their wider biological roles. While large scale screens aid the discovery of these peptides/microproteins in both physiology [149,150] and disease [22], a concerted effort towards their functional validation will help in ensuring they move from uncharacterized entities to mechanistically understood molecules. Experimental and computational approaches have begun to bridge this gap, enabling early predictions of peptide-coding transcripts. In silico experiments have enabled the prediction of DLEU1 as a peptide encoding lncRNA combined with even a prediction of its function as a water channel [151]. Moreover, Chao et al. (2022) transfected breast cancer cells with the lncRNA NDRG1-OT1 and reported the translation of a 66aa microprotein [152]. However, verifying the endogenous expression of such proteins is a critical next step. The sORFs should be validated through producing sORF-specific antibodies and detecting their presence in normal and cancer cells.

Expanding detection strategies, leveraging polysome profiling, and adopting new tools like machine learning in combination with established molecular biology techniques will be pivotal for the efficient identification and characterization of novel ncRNA-PEP/MPs, ultimately providing a clearer and more inclusive understanding of the proteome. Additionally, employing endogenous tagging when expression levels permit this offers a robust method for characterizing their functions.

Platforms like CPAT and CPC that predict the coding potential of transcripts often leverage factors like ORF size, ORF coverage, Fickett testcode, hexamer usage bias, and isoelectric points to predict RNA’s coding probability [153,154]. Many of the transcripts from our list of 86 are categorized as “non-coding” with such parameters. Such metrics could distinguish canonical and non-canonical transcripts efficiently, but whether they truly predict the coding potential of a transcript is worth looking into. Furthermore, some of the discussed peptides are encoded by bifunctional transcripts [44,155], which have both coding and non-coding functions. With a broader lens, certain mRNAs exhibit regulatory non-coding functions [156], making them bifunctional as well and truly blurring the line between coding and non-coding.

As we consider the potential of ncRNA-PEP/MPs in therapy and diagnostics, it becomes equally important to explore their evolutionary conservation and the implications for their broader biological significance. Many hypothesize that ncRNA-PEP/MPs exhibit limited evolutionary conservation; they are conserved among mammals [103] or even seen only among primates [88,94]. However, as a recently discovered class in the proteome, the “markers” defining such sequences remain poorly characterized, increasing the likelihood of underestimating homologs across species [144]. If they are indeed novel, their integration into existing pathways [36,71,114] highlights a fascinating interplay between newly emerging molecular players and established biological systems.

Addressing these gaps is critical for both advancing molecular biology and translating these discoveries into impactful clinical applications. The functional scope of ncRNA-PEP/MPs is incredibly complex, with many nuances still to be uncovered—a challenge that promises both excitement and discovery in the field.

## Figures and Tables

**Figure 1 cancers-17-01555-f001:**
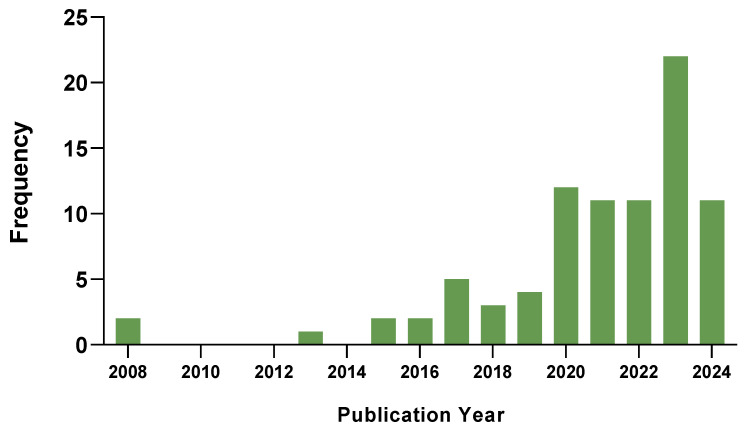
Publication dates for all ncRNA-encoded peptides retrieved from the literature search. In total, 78% of the papers we found that identify and characterize the function of ncRNA-PEPs (not reviews) are published 2020 onwards with 12 in 2020, 11 in 2021, 11 in 2022, 22 in 2023 and 11 in 2024 so far.

**Figure 2 cancers-17-01555-f002:**
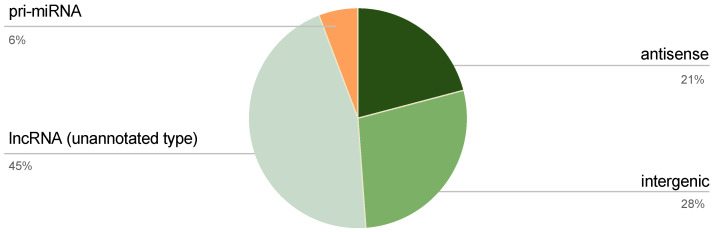
Classification of ncRNA transcripts that code for peptides. LncRNA-encoded peptides are shown in green, and those derived from shorter non-coding RNAs are represented in orange. Among the total ncRNA transcripts, only a portion (55%) was annotated by type. Peptides were encoded by 5 pri-miRNAs (6%), 18 antisense transcripts (21%), and 24 intergenic transcripts (28%), while 39 lncRNA transcripts (45%) remained unannotated.

**Figure 3 cancers-17-01555-f003:**
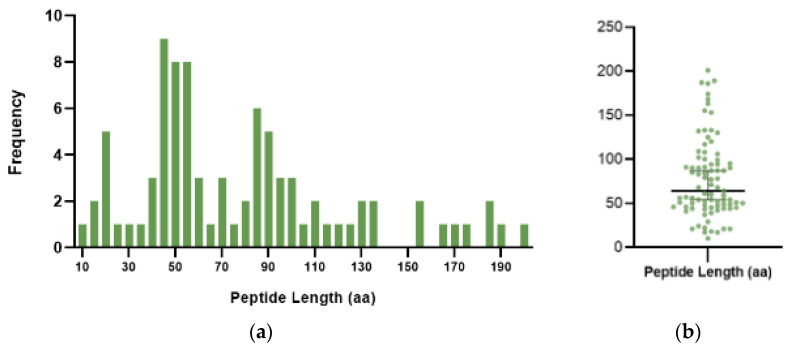
ncRNA-encoded product lengths were analyzed in bins of five amino acids to assess distribution patterns. (**a**) Histogram showing ncRNA-PEPs range from 10aa to 201aa and (**b**) box plot showing the median (64aa) and the 95% confidence interval.

**Figure 4 cancers-17-01555-f004:**
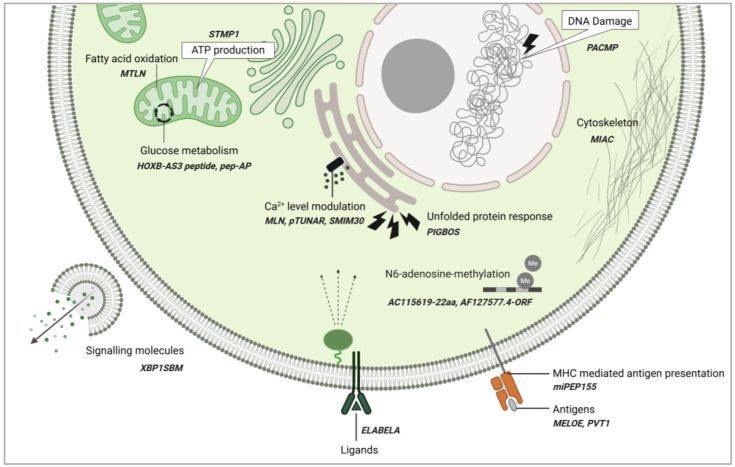
Illustration of the diverse cellular functions regulated by ncRNA-PEPs, highlighted in bold italics. MTLN, STMP1, pep-AP, and HOXB-AS3 peptides influence energy homeostasis. XBP1SBM upregulates VEGF expression, promoting angiogenesis. ELABELA functions as an endogenous ligand for the apelin receptor. In immune regulation, miPEP155 enhances antigen presentation, while MELOE peptides and PVT1 function as antigens themselves. MLN, pTUNAR, and SMIM30 modulate intracellular calcium levels by influencing the calcium ATPase SERCA. Under ER stress, PIGBOS participates in the unfolded protein response to maintain proteostasis and under genotoxic stress, PACMP supports DNA damage responses. AC115619-22aa and AF127577.4-ORF regulate N6-adenosine-methyltransferase assembly and stability, affecting co-transcriptional modifications. Finally, MIAC inhibits actin filament formation, shaping cytoskeletal integrity.

**Figure 5 cancers-17-01555-f005:**
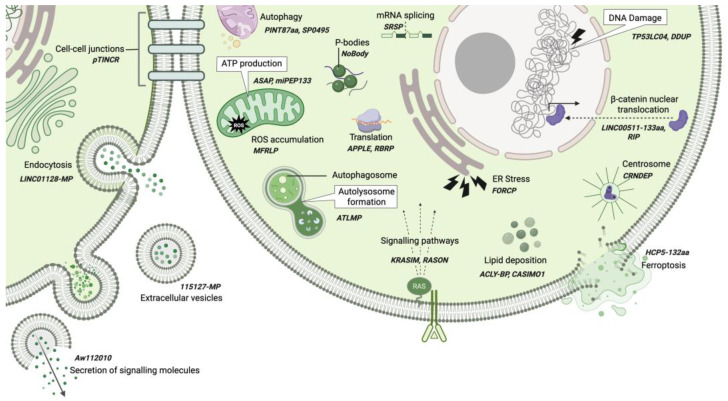
Figure Legend: Illustration of the diverse cellular functions regulated by ncRNA-MPs, highlighted in bold italics. In the mitochondria, ASAP and miPEP13 control ATP levels and MFRLP induces ROS accumulation. ATMLP, PINT87aa, and SP0495 participate in autophagy. SRSP and NoBody both participate in post-transcriptional modifications, by modulating mRNA splicing and p-body dissolution, ncRNA-MPs also play roles in translation regulation: RBRP modulates m6A recognition by the m6A reader IGF2BP1, and APPLE enhances translation initiation. They also influence signaling pathways: KRASIM and RASON modulate the MAPK pathway while RIP and LINC00511-133aa work in the Wnt pathway, controlling the nuclear translocation of β-catenin. Concerning lipid metabolism, CASMIO1 promotes lipid droplet clustering, and ACLY-BP stabilizes ATP citrate lyase (ACLY), leading to lipid deposition. Stress responses are also regulated by ncRNA-MPs, with FORCP managing ER stress and inhibiting apoptosis, HCP5-132aa preventing ferroptosis, and TP53LC04 along with DDUP participating in the DNA damage response. The expression of pTINCR induces the arrangement of cortical actin and strengthens cell-to-cell adhesion. LINC01128-MP participates in intracellular trafficking and endocytosis and CRNDEP associates with centrosomes during cell division. Aw112010 induces IL-12p40 production and 115127-microprotein is found in extracellular vesicles, suggesting roles in intercellular communication.

**Figure 6 cancers-17-01555-f006:**
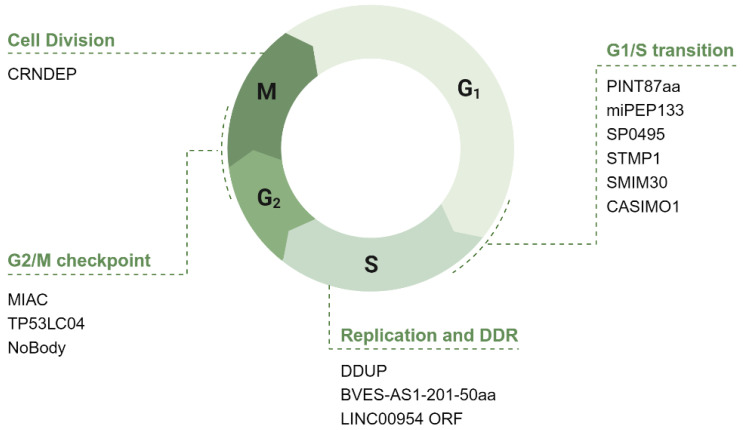
ncRNA-PEPs and ncRNA-MPs potentially regulate the progression of the cell cycle at different phases. PINT87aa, miPEP133, SP0495, STMP1, SMIM30, and CASIMO1 are involved in regulating the G_1_ phase or the G_1_/S transition in the cell cycle. DDUP, BVES-AS1-201-50aa, and LINC00954 are involved in the S phase while MIAC, TP53LC04, and NoBody participate in the G_2_/M transition. Finally, CRNDEP and Ub^KEKS^ show potential involvement the in the cell division phase of the cell cycle.

**Figure 7 cancers-17-01555-f007:**
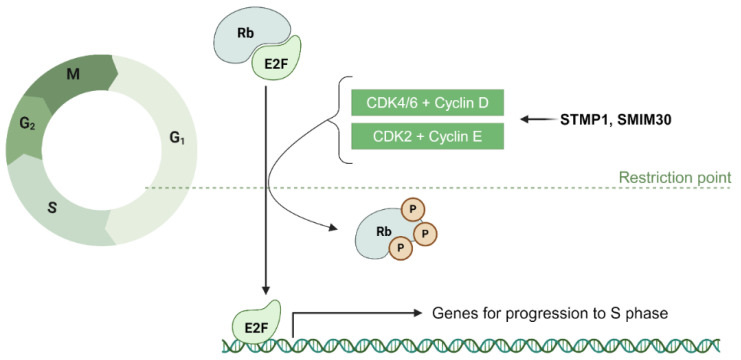
The Rb phosphorylation pathway is important at the restriction point. NcRNA-PEPs and ncRNA-MPs control different elements involved at this stage like transcription factors, CDKs and cyclins. In its hypo-phosphorylated state, Rb binds to E2F transcription factors, preventing them from activating genes necessary for DNA synthesis and cell cycle progression [123]. When cells receive signals to proliferate, cyclin D-CDK4/6 complexes become active and initiate the phosphorylation of Rb. This partial phosphorylation reduces Rb’s affinity for E2F. As the cell approaches the G_1_/S transition, cyclin E-CDK2 complexes further phosphorylate Rb, leading to its complete inactivation. NcRNA-PEPs STMP1 and SMIM30 are involved in modulating the CDK-cyclin complexes at this stage. The phosphorylation mediated inactivation of Rb fully releases E2F transcription factors, enabling the transcription of genes involved in DNA replication and S-phase entry.

**Figure 8 cancers-17-01555-f008:**
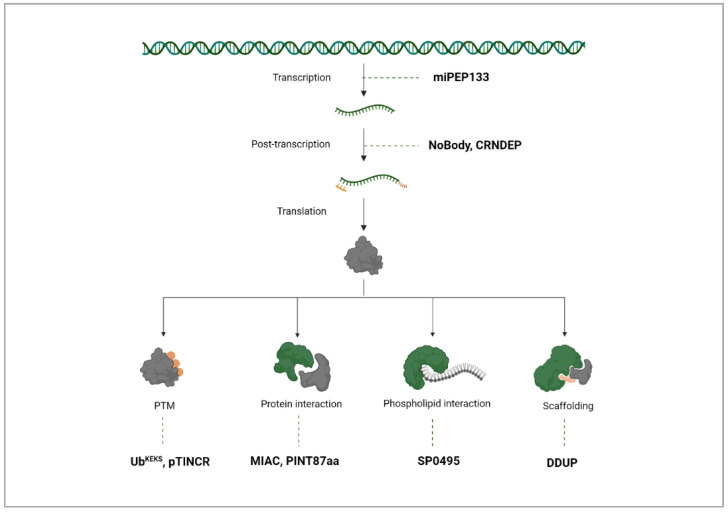
ncRNA-PEP/MPs involved in the cell cycle enforce their functions on many different levels. CRNDEP, miRNA-PEP133, and NoBody exert transcription and post-transcriptional control (PTM: post translational modifications). ncRNA-PEP/MPs are capable of interacting with other proteins to regulate them (MIAC, PINT87aa), modifying translated proteins (Ub^KEKS^, pTINCR) acting as scaffolds (DDUP) and binding phospholipids (SP0495).

**Figure 9 cancers-17-01555-f009:**
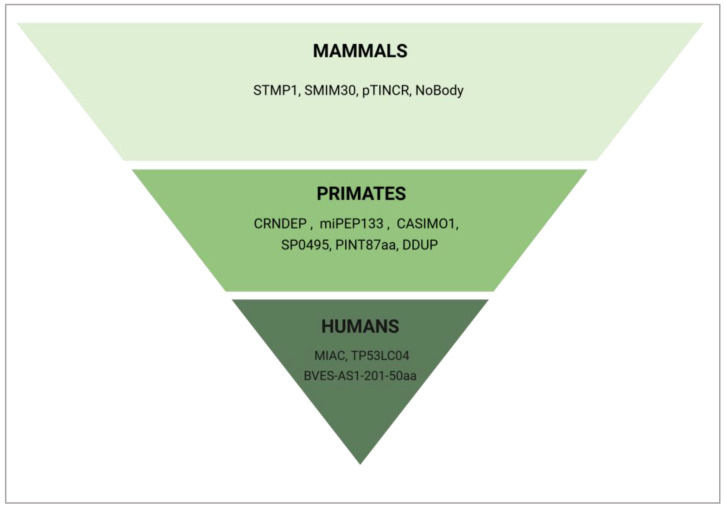
Conservation patterns of cell cycle-regulating ncRNA-PEP/MPs identified through BLAST analysis, filtered with an identity threshold of >85% and an E-value ≤ 1 × 10^−10^. The ncRNA-PEP/MPs discussed in this paper show conservation exclusively up to mammals, with no evidence of conservation in vertebrates or other lower taxa. Among these, STMP1, SMIM30, pTINCR, and NoBody are conserved within mammals, while CRNDEP, miRNA-PEP133, CASIMO1, SP0495, PINT87aa, and DDUP are restricted to primates. Interestingly, MIAC, BVES-AS1-201-50aa, and TP53LC04 exhibit conservation limited specifically to humans.

**Table 1 cancers-17-01555-t001:** List of strengths and limitations of detection methods for ncRNA-PEP/MPs.

Method	Strengths	Limitations
**Ribo-seq**	Maps ribosome-protected RNA fragments; identifies translation	Cannot distinguish functional vs. non-functional translation
**Mass spectrometry**	Direct peptide identification	Bias toward abundant proteins, missing low-expressed peptides
**Proteogenomics**	Integrates genomics with Mass spectrometry for novel peptide discovery	Computationally intensive
**Machine learning bioinformatics (e.g., sORFfinder, ORFscore)**	Predict sORFs and coding potential	Risk of false positives

**Table 4 cancers-17-01555-t004:** Cell cycle regulating ncRNA-PEPs and ncRNA-MPs.

ncRNA-PEP/MPs	ncRNA Transcript	Length (aa)	Cell Cycle Function	Sources
miPEP133	miR-34a	133	Induces mitochondrial dysfunction- p53 dependent and independent cell cycle arrest	[39]
PINT87aa	circPINT-exon2	87	G_1_ arrest, mitophagy inhibition via the PINK1/Parkin axis	[107]
SP0495	TP73-AS1/KIAA0495	201	G_1_ arrest, binds phosphoinositides, promotes autophagy	[114]
CASIMO1	lncRNA CASIMO1	83	Positively regulates mevalonate pathway, promotes G_1_/S transition	[18]
SMIM30	LINC00998	59	Controls intracellular calcium levels, modulates CDK4 and cyclin E2 to favor of G_1_/S transition	[70]
STMP1	lncRNA 1810058I24Rik	47	Enhances mitochondrial respiration, modulates cyclin E2, CDK2, and E2F1 to favor of G_1_/S transition	[75]
BVES-AS1-201-50aa	BVES-AS1	50	Promotes PCNA expression	[44]
DDUP	CTBP1-DT	186	Stabilizes Rad18 on damage sites and promotes homologous recombination and post-replication repair	[36]
LINC00954-ORF polypeptide	LINC00954	49	Downregulation of PCNA and CDK1	[51]
MIAC	AQP5-AS1	51	Binds AQP2 and controls EGFR signaling, overexpression causes arrest in S and G2 phases	[56]
TP53LC04	KLRK1-AS1	100	Arrests cells in response to DNA damage, part of the p53 response	[117]
NoBody	LINC01420	68	P-body dissolution during mitosis	[103]
CRNDEP	LINC00180	84	Centrosome maturation and microtubules	[19]
pTINCR	TINCR	87	Expression negatively correlated with cell cycle genes	[110]
Ub^KEKS^	UBBP4	76	Nuclear protein trafficking	[122]

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
