# Peer review of "Beyond the Transcript: Translating Non-Coding RNAs and Their Impact on Cellular Regulation"

_cancers, 2025, doi:10.3390/cancers17091555_

Round 1
Reviewer 1 Report
Comments and Suggestions for Authors
Review of the manuscript titled
“Beyond the Transcript: Translating Non-Coding RNAs and 2
Their Impact on Cellular Regulation”
by Ananya Deshpande et al.
This is a review article in which the authors focused on functionally characterized ncRNA-PEP and ncRNA-MP 74 and the mechanism of their functioning at the molecular level. Non-coding RNAs (ncRNA) constitute 98% of the human transcriptome. These molecules perform regulatory functions by interacting with DNA, RNA and proteins.
The article has been divided into several parts, which makes it very easy for the reader to provide the context of the entire manuscript.
The introduction characterizes the molecules in question, then describes sORFs, detection methods and mechanisms of translation of ncRNA ORFs.
The Authors described the role of ncRNA-PEP/MPs in cell cycle regulation, outlining the molecular pathways by which ncRNA-PEP and ncRNA-MPs can regulate cell cycle processes.
The Authors searched the scientific literature and identified 88 translations "non-coding" RNA. They proposed a new classification based on the length of amino acids, dividing their products into ncRNA-encoded peptides (ncRNA-PEP) or ncRNA-encoded microproteins (ncRNA-MP). Both peptides and microproteins play roles in health and disease by participating in the regulation of transcription and the formation of protein complexes.
Moreover, the article highlights progress in their detection, evolutionary origin, and potential applications in therapy.
This article systematizes knowledge about the role of ncRNA-PEP and
ncRNA-MPs as important regulators of cellular processes. Their functional versatility
is a challenge for further research in both cognitive and practical terms regarding applications in treatment. This review used a very rich literature containing 153 items that were correctly selected and cited in the text.
I have no critical comments. I believe that the article in its current form is eligible for publication.
Author Response
We thank the reviewer 1 for his/her comments.
The reviewer has no comments and provided following information.
"I have no critical comments. I believe that the article in its current form is eligible for publication."

Reviewer 2 Report
Comments and Suggestions for Authors
Comments on Review Manuscript: “Beyond the Transcript: Translating Non-Coding RNAs and Their Impact on Cellular Regulation”
This review is about the translation of non-coding RNAs (ncRNAs) into functional peptides and microproteins, challenging long-standing paradigms of ncRNA biology. This work contributes significantly to our understanding of ncRNA-derived translational products and their roles in cellular homeostasis, signaling, and disease, especially in the context of cell cycle regulation and cancer biology. Below are my comments and suggestions to enhance the manuscript.
- Remove the index content, please refer published work.
- Please define the cutoff length between peptides and microproteins.
- Give justify the distinction with examples where functional divergence exists between ncRNA-PEPs and ncRNA-MPs.
- In table 1, add column and transfer all ref to there
- Deepen the discussion on specific mechanisms by which ncRNA-PEPs or ncRNA-MPs interact with known complexes (e.g., CDK-cyclin complexes, PCNA, Rad18).
- Wherever please possible, provide experimental evidence (e.g., interactome studies, knockdown/knockout models) supporting these roles.
- Summarizing technological tools (e.g., Ribo-seq, proteogenomics, mass spectrometry, bioinformatic pipelines) in a comparative table is recommended.
- Discussing limitations and false-positive risks (e.g., translational noise vs. functional translation) should be there.
- Please cite examples of species-specific ncRNA-MPs that show conservation in function but not in sequence.
- Please highlight how evolutionarily novel microproteins might contribute to organismal complexity or disease susceptibility.
- Figure 4 and 5 should be coloured
- Ref this work and if important, include it. https://doi.org/10.1093/nar/gkad408, https://doi.org/11080/07853890.2024.2416604 , https://doi.org/10.3389/fcell.2024.1410914
Author Response
We thank the reviewer 2 for the comments.
Rebuttal for Reviewer 2
- Remove the index content, please refer published work.
The table of contents is removed and referred published work.
- Please define the cutoff length between peptides and microproteins.
In this review, we classify ncRNA-derived products as follows (please see the text under: Size Distribution of ncRNA-PEPs, lanes 220-221).
- Peptides: 1–60 amino acids (aa)
- Microproteins: 60–200 aa
- Proteins: >200 aa
This classification is based on emerging literature, where peptides are generally considered short, often disordered, functional molecules, while microproteins tend to form stable secondary structures, enabling complex interactions. For example, miPEP133 (133aa) encoded by pri-miR-34a, shows organized folding enabling interaction with mitochondrial chaperones, while SMIM30 (59aa) functions via relatively simpler membrane-associated mechanisms. We have added a sentence in the abstract to introduce the classification earlier in the text.
- Give justify the distinction with examples where functional divergence exists between ncRNA-PEPs and ncRNA-MPs.
The following text has been added and highlighted (with MELOE and PVT1 as examples) (Lanes 339-343).
Although ncRNA-PEPs and ncRNA-MPs are classified based on their length, a review of the literature suggests no clear functional divergence between the two. However, ncRNA-PEPs appear to have distinct roles in tumor immune evasion, particularly through the regulation of antigen presentation[55,68]. Given their strong association with anti-tumor immune responses, they are being proposed as potential candidates for cancer vaccines[120].
- In table 1, add column and transfer all ref to there
The references have now been moved to a new column for all tables.
- Deepen the discussion on specific mechanisms by which ncRNA-PEPs or ncRNA-MPs interact with known complexes (e.g., CDK-cyclin complexes, PCNA, Rad18).
Added and highlighted into the manuscript.
- Wherever please possible, provide experimental evidence (e.g., interactome studies, knockdown/knockout models) supporting these roles.
Following are sentences from the manuscript where experimental evidence has been added. We have added knockdowns, silencing, Co-IP and mass spec evidence. These data confirm biological relevance beyond ribosome occupancy or translation evidence.
- MIAC and PINT87aa have the ability to bind other proteins to modulate their functions, as confirmed by Co-IP experiments.
- The ncRNA-MP SP0495 has been shown to bind phosphoinositides, particularly PI(3)P and PI(3,5)P2, and to inhibit AKT phosphorylation[114] with protein-lipid overlay assays.
- Mass spectrometry analysis identified interactions between CRNDEP and proteins such as NuMA, Plk1, and Nlp, which are involved in centrosome maturation and mi-totic spindle formation, underscoring its potential involvement in mitotic regulation.
- Yang et al. (2023) find that silencing SMIM30 decreased the protein levels of CDK4, cyclin E2, and E2F1, as well as phosphorylated Rb.
- Frion et al. (2023) show that that UbKEKS knockout alters nucleolar protein composition, notably affecting proteins linked to cell cycle control and stress responses.
- This impairment of cell proliferation upon CASIMO1 knockdown in non-transformed MCF10a cells suggests that CASIMO1's function is relevant in physiological contexts as well.
- Knocking out PINT87aa in Hs683 glioma cells increased xenograft tumor volumes.
- Two ncRNA-MPs, PINT87aa and SP0495, induce G1 arrest upon overexpression[107,114].
- Furthermore, CRNDE(P) overexpression has been shown to enhance cellular re-sistance to various microtubule-targeting chemotherapeutic agents, such as paclitaxel and nocodazole, by accelerating microtubule polymerization and promoting cell sur-vival[19].
- Summarizing technological tools (e.g., Ribo-seq, proteogenomics, mass spectrometry, bioinformatic pipelines) in a comparative table is recommended.
A comparative table, as Table1, has been added:
|
Method |
Strengths |
Limitations |
|
Ribo-seq |
Maps ribosome-protected RNA fragments; identifies translation |
Cannot distinguish functional vs non-functional translation |
|
Mass spectrometry |
Direct peptide identification |
Bias toward abundant proteins, missing low-expressed peptides |
|
Proteogenomics |
Integrates genomics with MS for novel peptide discovery |
Computationally intensive |
|
Machine learning and bioinformatic tools (e.g., sORFfinder, ORFscore) |
Predict sORFs and coding potential |
Risk of false positives |
- Discussing limitations and false-positive risks (e.g., translational noise vs. functional translation) should be there.
The following paragraph has now been added and highlighted in the manuscript (lanes 125-140):
However, these techniques are not without limitations (Table 1). Ribosomes can occasionally translate non-functional sORFs, which may result in false positives when analyzing Ribo-seq data[25]. Each method also has inherent detection biases. Ribo-seq is more likely to detect abundant transcripts, while mass spectrometry often fails to identify peptides present at low abundance[9]. Expression data alone are insufficient to establish the biological relevance of a candidate sORF. Functional validation through assays such as gene knockout, overexpression, or interaction studies is essential to confirm a functional role. Moreover, bioinformatic prediction tools often identify a large number of ORFs that lack evolutionary conservation and show no evidence of cellular function, further complicating interpretation.[14].
Considering recent technical advancements and their limitations, it is essential to integrate computational tools, high-throughput screening, and classical molecular biology approaches to identify a previously underexplored class of molecules: translated ncRNAs and their corresponding peptides.
- Please cite examples of species-specific ncRNA-MPs that show conservation in function but not in sequence.
Functional conservation could persist despite the loss of sequence conservation if we look at proteins like pTINCR and DDUP that integrate into highly conserved pathways of cytoskeletal organization and DNA damage response, but they are likely fulfilling some unmet need in more complex organisms. However, the papers we have consulted have mainly looked at the sequence to assess conservation. In line with the reviewer comment, we cite papers (Patraquim, P., et al., 2022) that argue that the tools to study peptides and non-canonically translated proteins based only on sequence are not sufficient and highlight that new tools to address this very question are under development.
- Please highlight how evolutionarily novel microproteins might contribute to organismal complexity or disease susceptibility.
- Organismal complexity: Species-specific microproteins integrate into ancient pathways (e.g., STMP1 into mitochondrial regulation), adding modular control over conserved functions, enabling tissue-specific fine-tuning. Added a sentence in the evolutionary conservation section to highlight this point in the manuscript: This species-specific presence of ncRNA-PEP/MPs that interact with highly conserved proteins may be linked to organismal complexity.
- Disease susceptibility: If we look at the human specific proteins based on our BLAST criteria, MIAC and TP53LC04 appear to function as tumor suppressors, BVES-AS1-201-50aa has been characterized as oncogene. The report discusses their contribution to disease, but to the best of our knowledge, there is no clear evidence that evolutionary novelty itself is a common determinant of such susceptibility. These proteins are integrated into distinct cellular pathways and exhibit markedly different biological roles. Although it is plausible that recently evolved microproteins act as adaptive innovations that introduce regulatory flexibility at the potential cost of increased disease risk (like the primate-specific microprotein CASIMO1 which has been implicated in the regulation of lipid metabolism and cell cycle progression, potentially influencing breast tissue complexity and its associated pathologies), the current data do not provide a sufficiently strong foundation to support this as a general hypothesis.
- Figure 4 and 5 should be coloured
The figures have been adapted as per the comment.
- Ref this work and if important, include it. https://doi.org/10.1093/nar/gkad408, https://doi.org/11080/07853890.2024.2416604 https://doi.org/10.3389/fcell.2024.1410914
We have considered the sources above and decided not to include them as the references are not highly relevant to the review.
- https://doi.org/10.1093/nar/gkad408 is about the regulatory mechanisms of super enhancers and does not mention noncoding RNAs
- https://doi.org/11080/07853890.2024.2416604 we could not find the paper with this doi
- https://doi.org/10.3389/fcell.2024.1410914 outlines the mechanism of LncRNA Gm2044 functions in germ cell development. However, it does not encode a peptide or protein and thus is not relevant to this review

Reviewer 3 Report
Comments and Suggestions for Authors
This review focus on non-coding RNAs and their impact on cellular regulation. This is an excellent revision about this subject. Only a minor suggestion about the contextualization at the end pf the first section of the novelty of this review amount other related already published revision papers.
Author Response
We thank the reviewer 3 for the comments.
Rebuttal for Reviewer 3
- Contextualization at the end pf the first section of the novelty of this review amount other related already published revision papers.
Following paragraph has been added and highlighted in the manuscript:
Thus far, we have provided an up-to-date overview of ncRNA-derived peptides and microproteins and their diverse cellular functions. Building on this foundation, we now turn to a relatively underexplored aspect of the field—their potential involvement in cell cycle control. Given the central role of the cell cycle in development, tissue homeostasis, and disease, elucidating how these molecules influence its regulation is a critical and timely research direction. We summarize current findings, propose mechanistic links, and outline key gaps to inform future investigations.
